# Phage-inducible chromosomal islands promote genetic variability by blocking phage reproduction and protecting transductants from phage lysis

Rodrigo Ibarra-Chávez[1,2], Aisling Brady[1,3], John Chen[4], José R. Penadés[1,3,5]\*, Andreas F. Haag[1,6]\*

1 Institute of Infection, Immunity and Inflammation, College of Medical, Veterinary and Life Sciences, University of Glasgow, Glasgow, United Kingdom, 2 Department of Biology, Section of Microbiology, University of Copenhagen, Copenhagen, Denmark, 3 MRC Centre for Molecular Bacteriology and Infection, Imperial College London, United Kingdom, 4 Department of Microbiology and Immunology, Yong Loo Lin School of Medicine, National University of Singapore, Singapore, 5 Universidad CEU Cardenal Herrera, Moncada, Spain, 6 School of Medicine, University of St Andrews, North Haugh, St Andrews, United Kingdom

\* j.penades@imperial.ac.uk (JRP); afh22@st-andrews.ac.uk (AFH)

**Data Availability Statement:** All relevant data are within the manuscript and its Supporting Information files.

## Abstract

Phage-inducible chromosomal islands (PICIs) are a widespread family of highly mobile genetic elements that disseminate virulence and toxin genes among bacterial populations. Since their life cycle involves induction by helper phages, they are important players in phage evolution and ecology. PICIs can interfere with the lifecycle of their helper phages at different stages resulting frequently in reduced phage production after infection of a PICI-containing strain. Since phage defense systems have been recently shown to be beneficial for the acquisition of exogenous DNA via horizontal gene transfer, we hypothesized that PICIs could provide a similar benefit to their hosts and tested the impact of PICIs in recipient strains on host cell viability, phage propagation and transfer of genetic material. Here we report an important role for PICIs in bacterial evolution by promoting the survival of phage-mediated transductants of chromosomal or plasmid DNA. The presence of PICIs generates favorable conditions for population diversification and the inheritance of genetic material being transferred, such as antibiotic resistance and virulence genes. Our results show that by interfering with phage reproduction, PICIs can protect the bacterial population from phage attack, increasing the overall survival of the bacterial population as well as the transduced cells. Moreover, our results also demonstrate that PICIs reduce the frequency of lysogenization after temperate phage infection, creating a more genetically diverse bacterial population with increased bet-hedging opportunities to adapt to new niches. In summary, our results identify a new role for the PICIs and highlight them as important drivers of bacterial evolution.

**Funding:** This work was supported by the following grants awarded to JPR: MR/M003876/1, MR/S00940X/1 and MR/V000772/1 from the Medical Research Council (MRC, UK; https://mrc.ukri.org); BB/N002873/1, BB/S003835/1 and BB/V002376/1 from the Biotechnology and Biological Sciences Research Council (BBSRC, UK; https://bbsrc.ukri.org); 201531/Z/16/Z from the Wellcome Trust (https://wellcome.org). The funders had no role in study design, data collection and analysis, decision to publish, or preparation of the manuscript.

**Competing interests:** The authors have declared that no competing interests exist.

## Author summary

Bacteria need to protect themselves from infection and killing by phages to survive in the environment. For this purpose, bacteria have developed a sophisticated arsenal of defense mechanisms that can protect individual cells or the overall bacterial population. Individualized protection is achieved via systems such as CRISPR-Cas or lysogenization that allow the infected cell to survive. By contrast, population-based protection such as abortive infection systems lead to the cell's death before phage progeny is released. Here we describe a new role for phage-inducible chromosomal islands (PICIs) in protecting bacterial populations from phage predation. The resulting increased survival has consequences for the acquisition of foreign DNA such as antimicrobial resistance and fitness genes by the prey population as it also allows for the increased survival of bacteria that have acquired new genetic material. As a direct consequence of this increased survival, PICIs expand genetic diversity in bacterial populations. Such increased genetic diversity is advantageous to the complete bacterial population as the best adapted clones will outcompete others in any given environment. PICIs therefore also act as key mediators of population diversification.

## Introduction

Bacteriophages (phages) are viruses that infect bacteria and are estimated to be the most abundant biological entities on the planet [1]. Phages play an important role in the ecology of biological ecosystems and affect not only bacterial population structures, but also global carbon and nitrogen turnover [2,3]. They are likewise key mediators of horizontal gene transfer (HGT) via transduction, one of the most important processes driving bacterial evolution as well as the spread of antimicrobial resistance (AMR) genes. Transducing particles that contain bacterial DNA are generated during phage infection or prophage induction by a variety of mechanisms that lead to three main types of transduction: generalized, specialized, and lateral transduction (recently reviewed in [4]). Generalized transduction is the process by which phages package any bacterial DNA (chromosomal or plasmid) and transfer it to another cell. Here, the phage terminase "mistakenly" recognizes pseudo-*pac* sites located on the host chromosomal or on episomal DNA and package it into the phage particles [5,6]. Conversely, specialized transduction can package and transfer a limited segment of host bacterial DNA adjacent to the phage [7]. This type of transduction is the result of the erroneous excision of a prophage from the host chromosome. Consequently, in addition to the phage DNA, a small segment of prophage-adjacent chromosomal DNA is also packaged into the phage capsid resulting in transducing particles containing, in most cases, a defective phage. Specialized transduction has been described in both *cos*-type and *pac*-type phages [8]. During lateral transduction, replication and packaging of the prophage genome begins before its excision [9–11]. Thus, the phage packaging machinery can continue into the host chromosome leading to the generation of transducing particle that contain larger segments of chromosomal DNA adjacent to the prophage integration site. Since packaging is directional, high frequency transducing particles are only generated for chromosomal DNA to one side of the prophage. This mechanism has only been observed in *pac*-type phages because of the additional packaging constraints imposed on *cos*-type phages render it far less likely to occur.

 In the final stage of the lytic cycle, lysis of the host cells releases a mixture of phages and transducing particles, where phages significantly outnumber transducing particles. Once bacterial DNA is injected into a host cell, the transductants with the best chance of survival are

either those with lysogenic protection or those that rapidly acquire immunity against the surrounding phages. The last few years have seen a vast expansion of known phage defense mechanisms and studies have revealed that some of these are clustered in what is called defense islands [12]. These mechanisms frequently act at the nucleic acid level (i.e. restriction modification systems or CRISPR-Cas) or trigger host cell suicide after infection (abortive infection (Abi) systems) (see [13] for a recent review). Defense systems that can allow the host cell to survive phage infection are of particular interest as they can promote the survival and expansion of transduced bacterial clones. For example, lysogenization (by transducing phages) and CRISPR-Cas systems have been shown to increase the frequency of host cells transduced with bacterial DNA by protecting the transductants from phage lysis [14,15].

Phage-inducible chromosomal islands (PICIs) are a widespread family of phage satellites that parasitize helper phages for their own benefit [4,16]. After helper phage infection or prophage induction of a PICI-containing strain, PICIs are activated by a phage-encoded inducer and excise from the bacterial chromosome, replicate extensively, and are then packaged into capsids comprised of phage-encoded proteins [17,18]. The *Staphylococcus aureus* pathogenicity islands (SaPIs) were the first PICIs to be described [19], and are the prototypical members of this family of mobile genetic elements (MGEs). Importantly, PICIs can interfere with phage reproduction by hijacking the phage packaging machinery to promote packaging of their own genomes at the expense of phage transfer or interfering with phage late gene transcriptional regulation [20–23]. For example, the interference mechanisms of SaPI1 lead to a reduction in phage titer and an increased survival of bacteria relative to a SaPI1-free strain infected with a phage [19]. Preliminary observations in this study also indicated that infected SaPI-containing cells were less likely to result in phage lysogens and that this could be either due to abortive or failed phage infection [19]. All PICIs to date carry at least some of these interference mechanisms although the extent of interference caused by individual PICIs on their inducing phage varies considerably. These interference mechanisms not only ensure high frequency SaPI transfer, but also enable the formation of transducing particles that use the PICIs strategy for packaging [24]. While many PICIs carry genes with obvious selective advantages such as virulence factors and antibiotic resistance [25–27], some PICIs do not, suggesting that these islands could provide unexpected advantages to their host cells to persist in nature.

Based on the known ability of SaPIs to interfere with phage replication, we hypothesized that SaPIs, like other phage defense mechanisms [15,28,29], might benefit the bacterial population by allowing for the survival of an increased number of transductants after horizontal acquisition of bacterial DNA. Here we report that SaPIs promote the survival of bacterial cells after phage-mediated gene transfer by interfering with phage reproduction and reducing the population of infective phage particles. Our results indicate that this protection occurs at the population level and that this is enough to protect most of the bacterial cells from subsequent rounds of phage attack, allowing both cells that have and have not received foreign bacterial DNA to survive and persist in the population. Our work identifies a hitherto unexplored function for SaPIs in generating genetic diversity and shaping bacterial populations and highlights the role of this family of MGEs in bacterial evolution.

## Results

### SaPIs increase transduction

SaPIs are the prototypical members of the PICI family [17], and were used here as a model to investigate the impact of PICIs on phage-mediated transduction. Specifically, three different SaPIs were selected: (i) SaPIbov1 (GenBank accession number AF217235.1) and SaPI1 (GenBank accession number U93688.2) are clinically important since they encode menstrual toxic

shock syndrome toxin (TSST-1) and both have been extensively used for studying the mechanisms of PICI-phage induction, replication and phage interference; and (ii) SaPINY940, previously described as the incomplete phage pT1028, (GenBank accession number NC_007045.1) [30], which does not encode any identifiable toxins or virulence factors whose acquisition could be advantageous for its host. We further selected two temperate prophages (Φ11 and 80α) able to engage in generalized and lateral transduction to study the impact of each SaPI within a recipient strain on phage-mediated gene transfer.

To obtain transducing lysates, Φ11 or 80α lysogens carrying plasmid pJP2511 (Cm$^R$) or a chromosomal cadmium resistance cassette (Cad$^R$) next to the Φ11 or 80α *att*B sites were treated with mitomycin C (MC) for prophage induction [9]. Note that in these experiments the transfer of the plasmid (Cm$^R$) and the chromosomal (Cad$^R$) marker reflect generalized and lateral transduction, respectively [9,31]. Next, Φ11 lysates were used to infect (MOI 1:10) either the non-lysogenic, susceptible host strain, RN4220, or an RN4220 derivative carrying SaPIbov1, while 80α lysates were used to infect either RN4220 or RN4220 carrying either SaPINY940 or SaPI1 (see S1 Fig for an overview of the experimental setup). These combinations were chosen to generalize our results, and because Φ11 can induce SaPIbov1 [32], while 80α induces SaPINY940 and SaPI1. Note that SaPINY940 encodes a repressor identical to SaPI1, which is also induced by 80α [33]. Eighteen hours after infection, the different recipient cells were plated on selective media and the number of transductants were quantified. In support of our hypothesis that the presence of a SaPI in a recipient strain would be beneficial for the acquisition of foreign DNA by a bacterial population, the presence of either SaPIbov1, SaPI1 or SaPINY940 in the recipient host cells significantly increased the number of clones harboring the transduced Cad$^R$ or Cm$^R$ markers (Fig 1).

## Induction of the SaPI cycle is required to significantly increase transduction

In view of the previous results, we next analyzed whether the activation of the SaPI cycle in the recipient strain by the transducing phages was required to increase transduction. Interestingly, when we repeating the same experiments using Φ11 Δ*dut* or 80α Δ*sri* mutants, which do not encode the SaPIbov1 or SaPINY940/SaPI1 inducers, respectively [33], and therefore do not induce these SaPIs, the presence of either SaPIbov1 or SaPI1 did not increase transduction of the different markers (Fig 1). Unexpectedly, although the transduction frequencies observed for the 80α Δ*sri* mutant-mediated transfer of the different markers into the SaPINY940-positive strain were significantly lower than those observed when the wt phage (able to induce SaPINY940) was used, we still observed a minor increase in the transfer of the markers into the SaPINY940-positive strain, compared to the transfer into the SaPINY940-negative strain (Fig 1). This result suggested that contrary to what is seen with SaPI1 or SaPIbov1, the presence of SaPINY940 can provide a residual increase in the transfer of the markers independently of the activation of the SaPI cycle. Nevertheless and in concordance with what was seen with the other islands, the transfer of the marker was much more efficient when the SaPINY940 cycle was induced. The molecular basis of the different behavior among different SaPIs is currently under investigation.

Next, since SaPI induction severely interferes with phage reproduction [17], we hypothesized that a reduction in the phage titer after the first few rounds of infection led to an increase in the number of cells that survived later rounds of phage attack (including the cells transduced with bacterial DNA). In support of this hypothesis, the titers of the wt Φ11 and 80α phages, but not of the Φ11 Δ*dut* and 80α Δ*sri* mutants, were severely reduced after infection of the strains carrying SaPIbov1 or SaPI1 (Fig 2A). This reduction in phage titers was linked to a

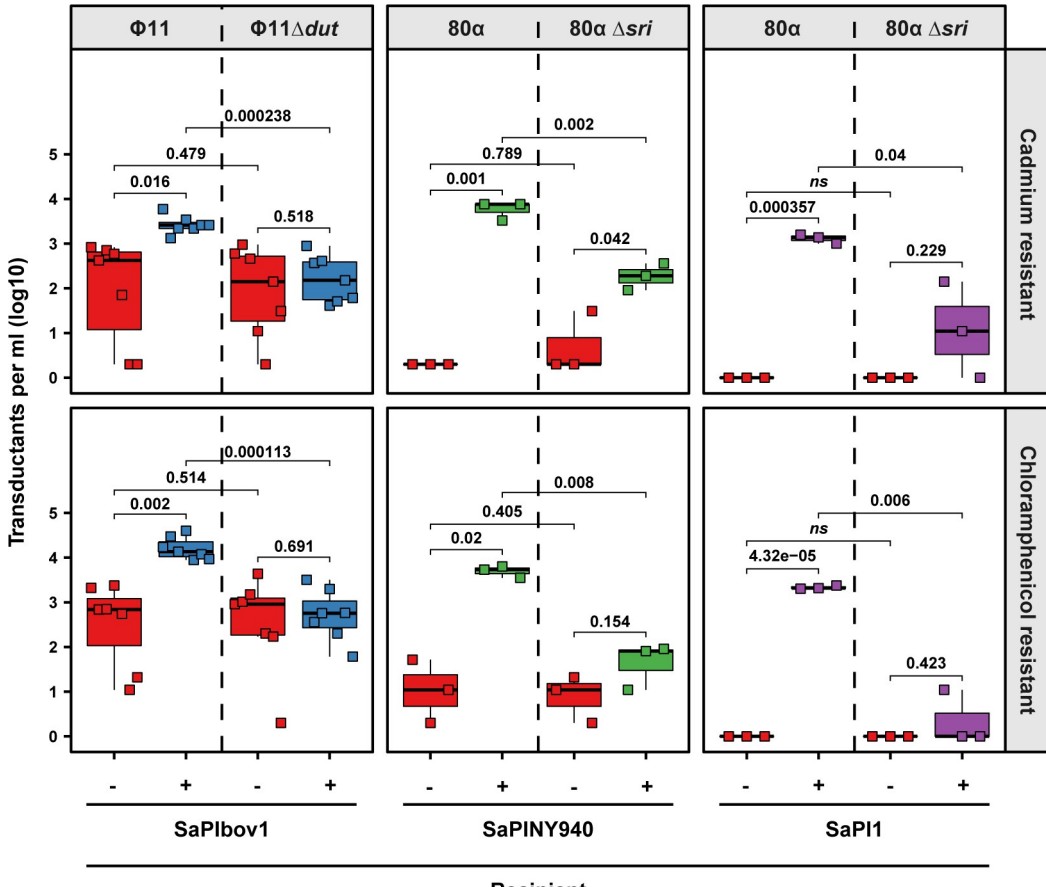

**Fig 1. SaPIs promote the survival of transductants.** *S. aureus* RN4220 derivative strains either devoid of (red) or harboring either SaPIbov1 (blue), SaPINY940 (green) or SaPI1 (purple) were infected with an MOI of 1:10 (phage:bacteria) of the indicated phages and acquisition of a chromosomal cadmium resistance marker (lateral transduction) (top row panels) or a plasmid-borne chloramphenicol resistance cassette (generalized transduction) (bottom row panels) was determined 18 h post-infection. Bold horizontal lines in each boxplot represent the median and lower and upper hinges the first and third quartiles, respectively (n = 7 biological replicates for strains infected with either Φ11 or Φ11 *Δdut* or n = 3 biological replicates for strains infected with either 80α or 80α *Δsri*). Assessment of statistically significant differences between groups was performed using a two-sided Student's t-test on log$_{10}$ transformed data. p-values are indicated above the respective comparison, ns not significant. Limit of detection (LOD) for this assay is 10 transductants per ml.

significant increase in the number of survivors present in the bacterial population (Fig 2B). In accordance with the increased transduction observed when the SaPINY940-positive strain was infected with the 80α *Δsri* mutant, a reduction in phage titer (Fig 2A) and an increase in cell viability (Fig 2B) were observed after infection of the SaPINY940-containing strain by phage 80α *Δsri*.

## SaPIs do not increase HGT but protect transductants from phage attack

To address whether SaPIs promoted increased levels of HGT or promoted the survival of transduced bacterial populations, we followed cell viability, phage titers and transduction titers of marker genes after 1, 4 or 18 h post infection (Fig 3). Note that the one-hour time point post infection allows for the completion of the initial transduction event but does not provide enough time for the release of phage progeny from this first infection event. We observed no significant differences in either viable cell counts, phage or transduction titers at this time.

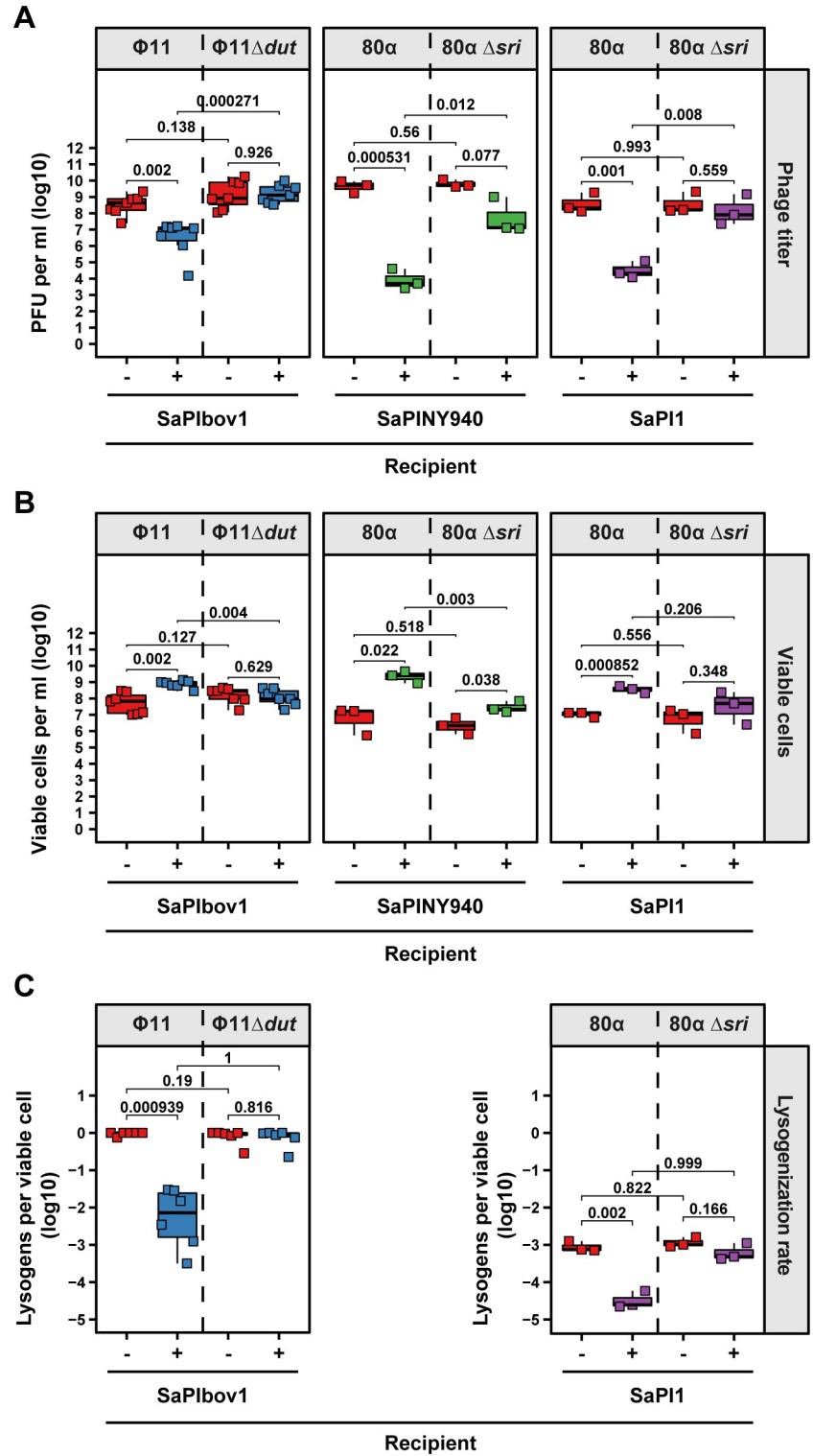

**Fig 2. SaPIs reduce phage reproduction and lysogenization but increase cell viability.** *S. aureus* RN4220 derivative strains either devoid of (red) or harboring either SaPIbov1 (blue), SaPINY940 (green) or SaPI1 (purple) were infected with an MOI of 1:10 (phage:bacteria) of the indicated phages. Phage titers **(A)**, viable cells **(B)** and the relative number of lysogens per viable cell **(C)** for each recipient were determined. Note that the number of lysogens could not be determined for SaPINY940 due to an incompatibility of selection markers. Bold horizontal lines in each boxplot represent the median and lower and upper hinges the first and third quartiles, respectively (n = 7 biological replicates

for strains infected with either Φ11 or Φ11 Δ*dut* or n = 3 biological replicates for strains infected with either 80α or 80α Δ*sri*). Assessment of statistically significant differences between groups was performed using a two-sided Student's t-test on $\log_{10}$ transformed data. p-values are indicated above the respective comparison, ns not significant. Limits of detection (LOD) for this assay are 100 plaques per ml **(A)**, 100 colonies per ml **(B)**.

This suggested that there was no inherent difference between the ability to acquire exogenous DNA via HGT in either SaPI-positive or SaPI-negative strains. However, differences began to be evident 4 h after the phage lysates were added to the recipients: SaPI-containing cells showed an increased number of viable cells and transductants as well as reduced phage titers relative to the SaPI-free recipient strains or strains infected with the phage inducer mutants (Fig 3). To estimate whether bacterial population underwent additional transduction events after the initial infection cycle, we calculated the ratio of transduced to viable cells at 1 h and 18 h post infection. In cases where subsequent transduction events occurred, this number should increase at the later timepoint, reflecting a larger portion of the bacterial population to have acquired the relevant resistance markers. We did not observe any increase in this ratio, further suggesting that transduction of the marker genes occurred during the first infection cycle and did not play a major role during later timepoints.

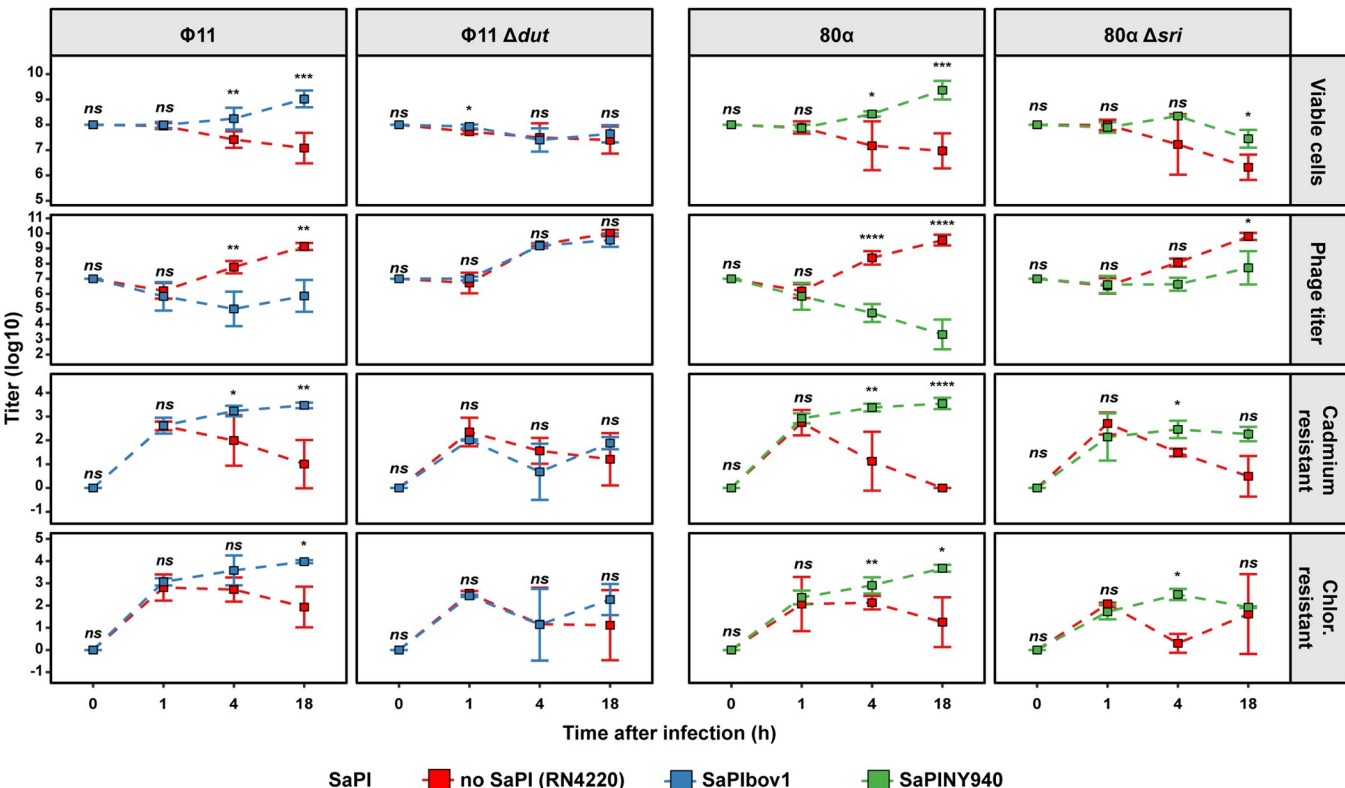

**Fig 3. Assessment of strain viability, phage titers and transduced markers at different time points following infection.** The indicated strains either devoid of (red) or harboring either SaPIbov1 (blue) or SaPINY940 (green) were infected with an MOI of 1:10 (phage:bacteria) with the indicated phage donor lysates. Samples for viable cell count, phage titer and number of cells transduced with either antibiotic resistance marker were taken at the indicated timepoints. Note that at 0 h, viable cells correspond to the number of infected cells used, phage titers correspond to the number phages from the donor lysate added for the infection and resistance marker titers are 0 as the recipient cells are sensitive at this stage. Error bars are s.d. from the mean. Assessment of statistically significant differences between groups was performed using a two-sided Student's t-test on $\log_{10}$ transformed data. p-values are as follows: * p<0.05, ** p<0.01, *** p<0.001, **** p<0.0001, ns not significant. Limits of detection (LOD) are 100 colonies per ml for viable cells counts, 100 plaques per ml for phage titers and 10 colonies per ml for transductions.

## SaPIs protect bacterial populations through a range of MOIs

To test whether SaPIs could mediate population protection when exposed to higher initial phage burdens, we repeated the infection experiment using an MOI of 1:1 rather than the MOI of 1:10 that was used in the previous experiments (Fig 4). This high MOI was selected to ensure that most bacterial cells would undergo a phage infection (~63% at an MOI of 1:1 compared to ~9.5% at an MOI of 1:10 assuming Poisson distribution of infection) as well as to balance the drive of temperate bacteriophages to lysogenize at higher MOIs [34,35]. Even when

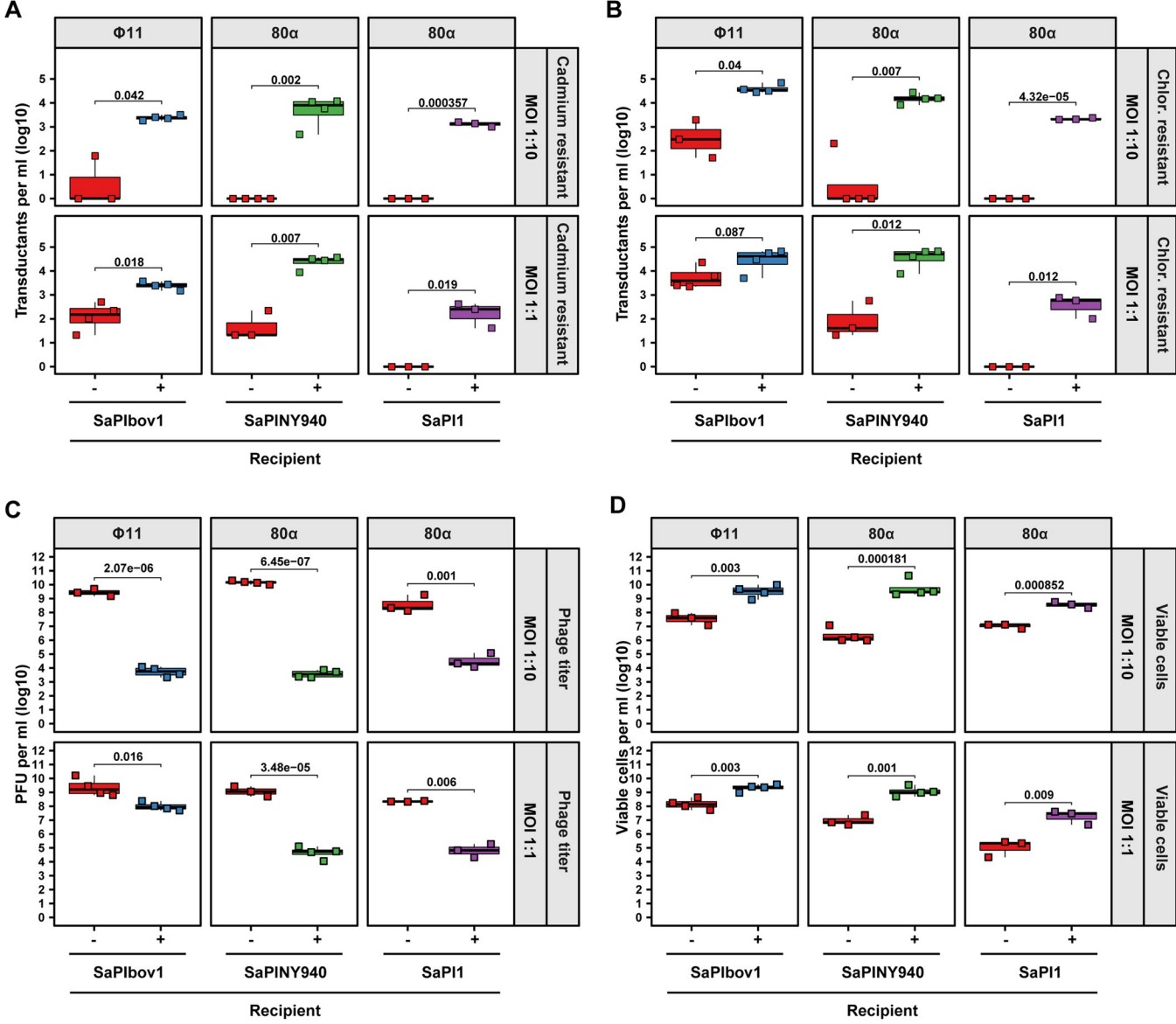

**Fig 4. Impact of MOI on horizontal gene transfer, phage titers and viability.** *S. aureus* RN4220 derivative strains either devoid of (red) or harboring either SaPIbov1 (blue), SaPINY940 (green) or SaPI1 (purple) were infected with an MOI of either 1:1 or 1:10 (phage:bacteria) of the indicated phages and acquisition of a chromosomal cadmium resistance marker (lateral transduction) **(A)**, a plasmid-borne chloramphenicol resistance cassette (generalized transduction) **(B)**, phage titers **(C)** and cell viability **(D)** was determined 18 h post-infection. Bold horizontal lines in each boxplot represent the median and lower and upper hinges the first and third quartiles, respectively (n = 3–4 biological replicates as indicated). Assessment of statistically significant differences between groups was performed using a two-sided Student's t-test on $\log_{10}$ transformed data. p-values are indicated above the respective comparison, ns not significant. Limit of detection (LOD) of transductants **(A&B)** is 10 transductants per ml, 100 plaques per ml for phage titers **(C)**, and 100 colonies per ml for viable cells **(D)**.

cells were infected with 10-fold the number of phages, SaPI-positive cells retained an increased number of transductants, a reduced phage titer and increased viability compared to their SaPI-negative relative (Fig 4). These differences were more robustly maintained in experimental systems involving phage 80α compared to the system with Φ11, indicating that the afforded protection was likely dependent on the phage-SaPI pair as well as the infectious dose.

## SaPIs increase genetic diversity in the recipient cells by reducing lysogenization

Prophages are known to promote transduction by protecting recipient cells from phage attack, preventing further superinfection by the same phage type (superinfection immunity) [36]. Consequently, most of the surviving transductants become lysogenic for the transducing phage in the absence of a PICI within the recipient population [14]. The fact that SaPIs modify phage dynamics prompted us to study the effect that SaPIs may have on the frequency of lysogeny after phage infection. To do that, we took advantage of the fact that the wt and mutant Φ11 and 80α prophages carried an erythromycin resistance marker *erm*C integrated into their phage genomes [37]. The lysates obtained in Fig 2A were used to infect either RN4220 (susceptible strain, Φ11 and 80α lysates) or RN4220 carrying either SaPIbov1 (Φ11 lysates) or SaPI1 (80α lysates), (MOI 1:10). Note that SaPINY940 was not used in these experiments because it carries the same *erm*C marker present in the phages. Eighteen hours after infection, the bacterial cells that survived the phage attack were grown on TSA plates lacking or containing erythromycin, and the proportion of the lysogens (compared to the viable cells) quantified. The SaPIbov1- and SaPI1-containing strains showed a significantly reduced proportion of cells becoming lysogenic compared to the susceptible RN4220 strain when infected with either wt Φ11 or wt 80α, respectively. By contrast, no differences between the SaPI-containing strains and RN4220 were observed when infecting them with either their respective inducer mutant lysates (Φ11 Δ*dut* for SaPIbov1 and 80α Δ*dut* for SaPI1) (Fig 2C).

The previous results suggested that the presence of the island will not only increase the survival of transductants but also the genetic diversity of the bacterial population by preserving part of the original non-lysogenic population and promoting the existence of additional different cell types, carrying different genetic information. To test this idea, we introduced into the lysogenic strains for Φ11 (wt or Δ*dut*) both the chromosomally encoded Cd marker, and the plasmid pJP2511, which encodes Cm^R. These strains were MC induced, and the lysates were used to infect strain RN4220 or its SaPIbov1-containing derivative. From this experiment we determined the number of viable cells, lysogens, transductants as well as number of transductants becoming lysogenic by replica-plating them onto TSA supplemented with erythromycin. These data were then used to further assess the impact of SaPIs in a host cell on population diversity, by calculating the relative frequencies of different cell populations as a proportion of viable cells within each replicate. In support of the hypothesis, and as is shown in Fig 5, in absence of SaPIbov1, all the cells became lysogenic for Φ11 (Fig 5C, orange population, Erm^R), which implied that the original non-lysogenic RN4220 strain (Fig 5C, grey population) was no longer present in the population. Moreover, in absence of SaPIbov1, two additional cell types appeared in the population, which corresponded to derivative lysogenic cells for Φ11 carrying either the chromosomally encoded Cm^R marker or the pJP2511 plasmid (Fig 5C, depicted in yellow). By contrast, the presence of SaPIbov1 notably increased diversity in the population, not only by modifying the frequency distribution of the aforementioned clones generated in absence of the island, but also by promoting the existence of three additional cell types (Fig 5). The first one corresponded to the original non-lysogenic RN4220 (grey population), which is absent when the island is not present but remained as the most prevalent clone in the

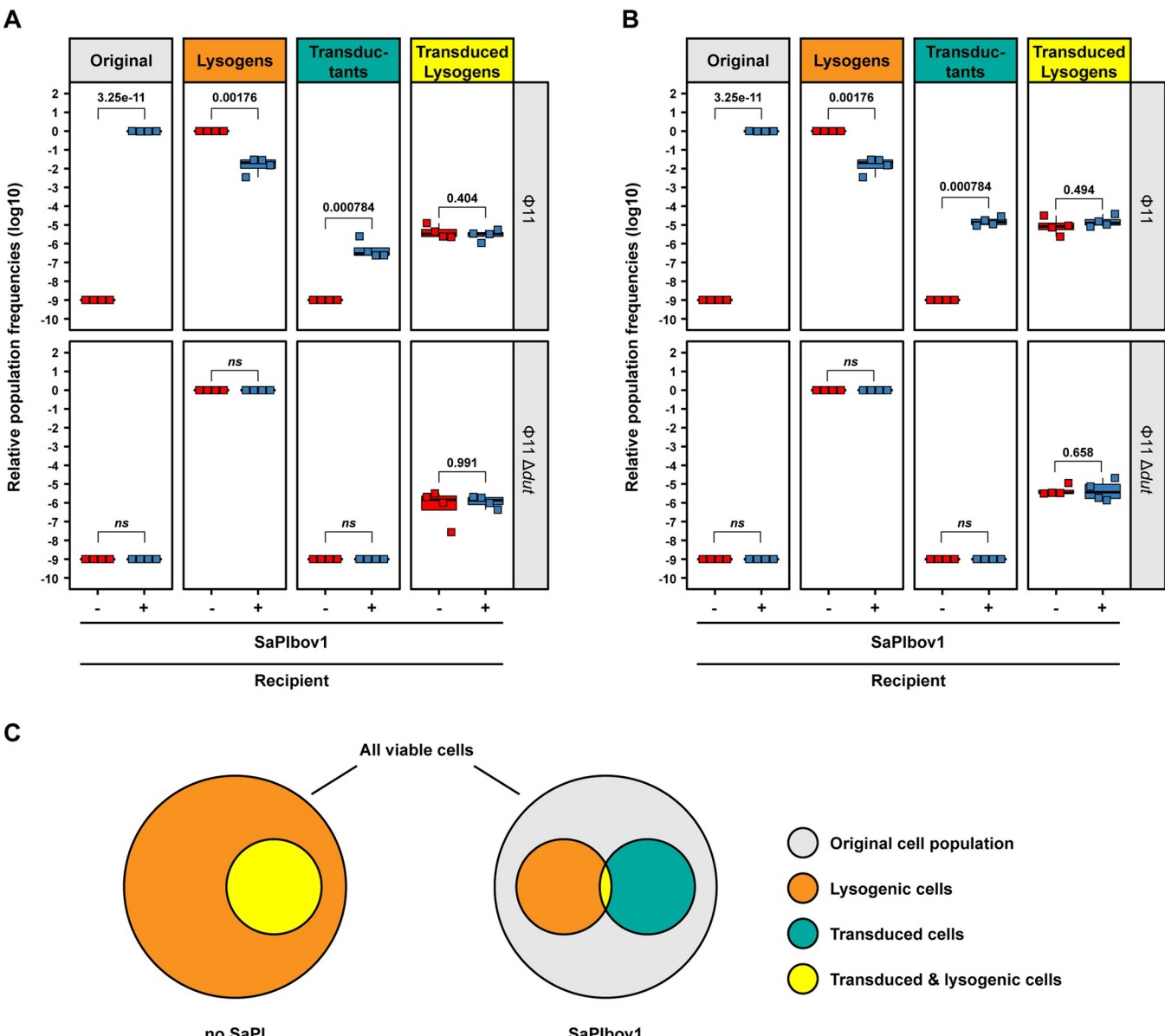

**Fig 5. SaPIs increase population heterogeneity.** *S. aureus* RN4220 derivative strains either devoid of (RN4220) or harboring SaPIbov1 were infected with an MOI of 1:10 (phage:bacteria) of a Φ11 or Φ11 Δ*dut* mutant lysate derived from a donor strains containing a chromosomal cadmium and plasmid-borne chloramphenicol resistance marker. Number of viable cells, transductants and lysogenization frequencies of the defined populations and their relative frequencies to the total number of viable cells were determined. Population frequencies of **(A)** cells acquiring the chromosomal cadmium resistance marker or **(B)** the plasmid-encoded chloramphenicol resistance marker in the defined recipient strains. Bold horizontal lines in each boxplot represent the median and lower and upper hinges the first and third quartiles, respectively (n = 4 biological replicates for all samples). Assessment of statistically significant differences between groups was performed using a two-sided Student's t-test on log$_{10}$ transformed data. p-values are indicated above the respective comparison, ns not significant. Note that $10^{-9}$ as relative population frequency in these data indicates populations that were completely absent. **(C)** Schematic representation of the presence of different cell populations in either a recipient lacking a SaPI (left) or containing a SaPI (right).

population when the island is present. The other two clones were derivative RN4220 strains carrying the antibiotic resistance markers, but not the prophage (Fig 5, depicted in green). Of note is that the proportion of cells that had both become lysogenic for Φ11 and acquired one of the transduced resistance markers showed no differences between either the SaPI-positive or the susceptible RN4220 strain, suggesting that in the generation of this subpopulation (i)

transduction and lysogenization were coupled, (ii) had to occur during early infection and (iii) were independent of the presence of a SaPI and other cell populations. Taken together, these results confirmed an unexpected role of the SaPIs in bacterial evolution by promoting genetic variability.

## Discussion

Transduction is a fundamental process of bacterial evolution that hinges on the survival of the transduced cells (from further phage attack) in order to persist. By protecting their host cells from phage killing, phage defense mechanisms such as lysogenic protection or CRISPR-Cas, increase the chances of any given cell's survival within the population while Abi-systems act on the whole bacterial population by preventing the release of phage progeny. Here we show that PICIs are not merely selfish MGEs but provide additional benefits to their host cells by blocking phage reproduction and thus promoting the survival of the entire bacterial population, including those that have acquired foreign genetic material (Fig 6). While prophages and CRISPR-Cas systems provide defense at the single cell level, PICIs are likely operating at population level through a general reduction in phage titers. While PICI-containing cells show no differences in phage infection, phage burst sizes are known to be reduced in PICI-containing cells that are unsuccessful in fully blocking the infecting phage [19]. Whether PICIs can also contribute to protecting individual cells from phage predation remains to be determined but is likely to be dependent on the individual phage-PICI combination. Untangling the protective effects that are eventually attributable to PICIs or simply the result of lysogenic protection (particularly at high MOIs), could provide further insights into the evolutionary impacts of these versatile molecular parasites. In this study, protection of host populations is dependent on the induction of the PICIs following host cell infection in all three PICIs tested. However, we cannot discard the possibility that some PICIs may confer protection even in the absence of phage induction. Indeed, some evidence for this can be gleaned from our results when infecting a strain containing SaPINY940 with a phage unable to induce this island. Even though a substantial amount of protection is lost in absence of island replication, there was still evidence of an increased level of protection in this strain background compared to its PICI-free relative. While this observation will require further confirmation, we anticipate PICI mediated protection might also act against non-inducing phages. This is currently under investigation.

Another interesting observation of our study is that SaPIs can substantially reduce the proportion of phage lysogens in the bacterial population which results in a more heterogenous population overall. Since lysogenization is a stochastic process that occurs randomly [38–40] and is promoted by high MOIs [34,35], we initially hypothesized that this reduction in the number of lysogens might simply reflect the reduction in phage titers caused by the SaPI interference. However, the consistent protection of the bacterial population and the resulting reduction in phage titers even at higher MOIs suggest that SaPIs might be more apt at blocking infecting phages than previously appreciated. In any case, such genetic heterogeneity can afford the bacteria with bet-hedging opportunities that enable them to better adapt and survive in changing and stressful environments.

While it is true that the horizontal transfer of specific genes can be beneficial to a cell if the acquired genes provide an advantage to the recipient cells compared to the rest of the population, it can also be detrimental if the mobilized genes have no function or are incompatible with existing genes [41]. The same applies to lysogenization, which can provide positive or negative effects, depending on the environmental context [42]. Importantly, our results demonstrate that SaPIbov1 increases the genetic variability of the recipient cells, not just by preserving the original lineage but also by creating new lineages that do not appear when the

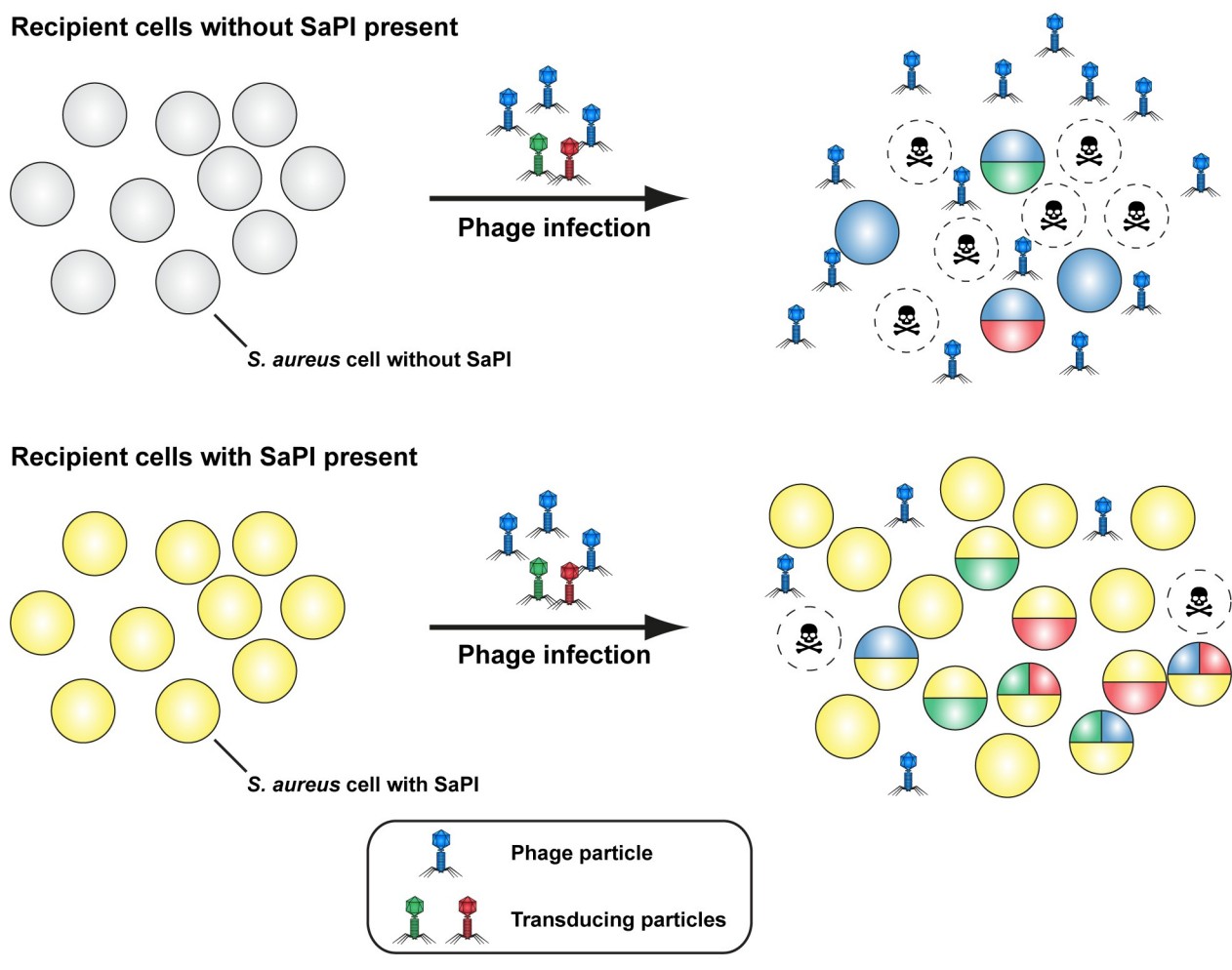

**Fig 6. Model of the impact of SaPI protection from phage predation on horizontal gene transfer and population heterogeneity.** In SaPI-free recipients (grey cells), infecting phages can reproduce without restrictions and lyse their host cell (indicated dashed lines and skull and crossbones symbol). Only lysogenization (cells shaded blue) can protect the host cell from future phage infection and killing. Any cells with horizontally acquired genes (red and green shaded cells) that do not undergo lysogenization are also subject to subsequent rounds of phage infection increasing the likelihood of their loss. In the presence of SaPIs, recipient strains can successfully block phage replication and can survive without the need for lysogenization. Consequently, transductants are also more likely to survive leading to an overall increase in population heterogeneity. We propose that this occurs mainly via a general reduction of phage reproduction and burst size protects at population level. However, SaPIs could also act at the individual cell level and protect the infected cell from lysis. The mechanistic details of protection are likely to be diverse and different SaPIs could employ replication-dependent and -independent mechanisms to achieve this.

island is not present, highlighting how these events can further influence the evolution and clonal expansion of bacteria. Since the output of the transfer can be positive or negative for the recipient cells, PICIs can therefore increase the chance of the bacterial population to survive by minimizing the negative effects that the horizontal transfer of undesirable genes may have in the recipient cells by promoting genetic diversity. In essence, genetic variability is facilitated by the presence of PICIs in a host strain and natural selection will determine the fittest clone.

In nature, bacteria live in polymicrobial communities that contain abundant phage loads [43]. These phages are not only immensely important in shaping the bacterial community but are a major hub for the inter- and intrageneric exchange of genetic information [44]. Among the genetic material on offer are virulence and fitness factors as well as antimicrobial resistance genes. By increasing the frequency of genetic material exchange via increased survival of trans-duced clones, SaPIs can no longer be considered as self-serving MGEs but rather as major

contributors to bacterial evolution. SaPIs benefit the host population by effectively acting as a host defense strategy, undermining phage reproduction, and increasing the chances of acquiring new sets of genes by transduction [45]. The fact that PICIs are found across distantly related taxa suggests that their lifestyle has a strong selective value representing a novel strategy for conferring fitness and promoting genomic variability among bacteria and adds another defense element to the mobile bacterial immune systems [29]. Following this rationale, the protective role of SaPIs, and other PICIs could potentially affect the application of phages as a means to reshaping microbiomes and eliminate specific pathogenic strains. This aspect should be evaluated in phage therapy and gene delivery to avoid the unintended fitness enhancement by transduction which could cause potential risks associated with public health and antimicrobial resistance.

## Materials and methods

### Bacterial strain and culture conditions

The bacterial strains and plasmids used in this study are detailed in Table 1. *S. aureus* strains were grown in Tryptic soy broth (TSB) or on Tryptic soy agar (TSA) plates. Antibiotic selection was used where appropriate (erythromycin 10 μg ml$^{-1}$, chloramphenicol 20 μg ml$^{-1}$, cadmium chloride 100 μM).

**Table 1. Strains and plasmids used in this study.**

| Strain | Description | Reference |
|---|---|---|
| JP1996 | RN4220 SaPIbov1 *tst*::*tet*M | [46] |
| JP4125 | RN451 Φ11 Δ*dut* | [33] |
| JP6022 | RN10359 80α Δ*sri* | [33] |
| JP6399 | RN4420 lysogenic for 80α harboring an erythromycin resistance cassette | [47] |
| JP6400 | RN4220 lysogenic for Φ11 harboring an erythromycin resistance cassette | [48] |
| JP14277 | RN4220 SAOUHSC_01121::*cad*CA; cadmium resistance cassette inserted 35 kb downstream of Sa7 *att*B | [9] |
| JP19047 | RN4220 harboring pT1028 marked with and erythromycin resistance cassette | [49] |
| JP19145 | RN4220 SAOUHSC_01121::*cad*CA; cadmium resistance cassette inserted 5 kb downstream Sa5 *att*B | [9] |
| JP20714 | JP14277 pJP2511 | This study |
| JP20716 | JP19145 pJP2511 | " |
| JP20718 | JP6022 with transduced with a lysate of JP6399 *erm*C (80α Δ*sri* harboring an erythromycin resistance cassette) | " |
| JP20722 | JP4125 transduced with a lysate of JP6400 resulting in Φ11 Δ*dut* harboring an erythromycin resistance cassette | " |
| JP20844 | JP20714 lysogenic for 80α harboring an erythromycin resistance cassette | " |
| JP20845 | JP20714 80α Δ*sri* harboring an erythromycin resistance cassette | " |
| JP20846 | JP20716 lysogenic for Φ11 harboring an erythromycin resistance cassette | " |
| JP20847 | JP20716 lysogenic for Φ11 Δ*dut* harboring an erythromycin resistance cassette | " |
| RN10359 | RN450 lysogenic for 80α | [50] |
| RN10616 | RN4220 lysogenic for 80α | [50] |
| RN10822 | RN4220 SaPI1 *tst*::*tet*M | [51] |
| RN450 | NCTC8325 cured of Φ11, Φ12 and Φ13 | [52] |
| RN451 | RN4510 lysogenic for Φ11 | [50,52] |
| RN4220 | Restriction-defective derivate of RN450 | [53] |
| **Plasmids** | **Description** | **Reference** |
| pJP2511 | Gram-positive plasmid containing a chloramphenicol resistance cassette, Cm$^R$ | [54] |

## Phage induction and titration

*S. aureus* strains lysogenic for the defined phages and containing the defined chromosomal markers or plasmids were grown to early exponential phase (OD$_{540}$~0.15) at 37˚C and 120 rpm. Cultures were then induced by the addition of mitomycin C (2 μg ml$^{-1}$) and incubated for 4–5 h at 30˚C followed by overnight incubation at room temperature before filtering with a 0.2 μm syringe filter (Sartorius). To determine the phage titers, RN4220 cultures were grown to OD$_{540}$~0.35 and 100 μl of this culture was mixed with 3 ml of phage top agar (PTA, 20 g l$^{-1}$ Nutrient Broth No. 2, Oxoid, plus 3.5 g l$^{-1}$ agar, Formedium supplemented with 10 mM CaCl$_2$) and overlaid onto phage base agar plates (20 g l$^{-1}$ Nutrient Broth No. 2, Oxoid, plus 7 g l$^{-1}$ agar, Formedium supplemented with 10 mM CaCl$_2$). Phage lysates and dilutions in phage buffer (PHB) (1 mM MgSO$_4$, 4 mM CaCl$_2$, 50 mM Tris-Cl, 100 mM NaCl, pH = 8) were spotted in triplicates of 10 μl each onto lawns of the specified strains, dried and incubated overnight prior to plaque forming unit (PFU ml$^{-1}$) determination.

## Phage time-course infection experiment

Cultures of the indicated strains were grown to exponential phase and normalized to an OD$_{540}$ of 0.25 corresponding to ~1 x 10$^8$ CFU ml$^1$. Five to ten ml of this suspension were then supplemented with 5 μM CaCl$_2$ and infected with the defined phage lysates at an MOI of either 1:10 (phage:bacteria) and incubated at 30˚C and 80 rpm. At the indicated timepoints, samples were taken to assess the number of viable cells, transductants and phage titers.

## Viable cell count determination

Cultures were serially diluted in PBS and spotted in triplicates of 10 μl per dilution onto a TSA plate. The plates were incubated for 18 h and the colony forming units determined.

## Transductions of resistance markers

At the defined timepoints, 100 μl of culture was plated directly onto TSA supplemented with the appropriate antibiotics. Plates were then incubated for 24–48 h (depending on the resistance marker assessed) and colonies were counted for transduction titer enumeration (TFU ml$^1$).

## Phage lysogenization

Lysogenization of Φ11 was assessed either directly from culture or by replica plating surviving colonies or transductants onto TSA plates supplemented with 10 μg ml$^{-1}$ erythromycin, as this phage contained an erythromycin resistance cassette within its genome.

## Statistical analysis

Statistical analysis and plotting was performed using RStudio version 1.4.1717 and R version 4.1.0 [55] with the following packages installed: ggplot2 [56], tidyverse [57], stringi [58], plyr [59], dplyr [60], rstatix [61] and datarium [62]. Details are annotated in the relevant figure legends.

## Supporting information

**S1 Fig. Experimental set up.** Recipient cultures were grown to exponential phase (OD$_{540}$ of 0.25, corresponding to ~1 x 10$^8$ CFU ml$^1$) in TSB. 5 ml of each recipient culture supplemented with 5 μM CaCl$_2$ were infected with indicated phage lysates at an MOI of 1:10 (phage:bacteria).

Cultures were incubated at 30°C and 80 rpm and samplings were performed at the indicated timepoints (1, 4 and 18 h). For transduction assessment, 100 μl of each culture at the defined timepoints were plated directly onto TSA plates supplemented with the appropriate antibiotics. For viability assessment, cultures were serially diluted in PBS and spotted in triplicates of 10 μl per dilution onto TSA plates. For phage titer assessment, 1 ml of each culture was filtered and used for serial dilutions in phage buffer (PHB). Dilutions were spotted in triplicates of 10 μl each onto PBA plates overlaid with a lawn of RN4220. Created with BioRender.com. (TIF)

## Author Contributions

**Conceptualization:** José R. Penadés, Andreas F. Haag.

**Formal analysis:** Rodrigo Ibarra-Chávez, Aisling Brady, John Chen, José R. Penadés, Andreas F. Haag.

**Funding acquisition:** José R. Penadés.

**Investigation:** Rodrigo Ibarra-Chávez, Aisling Brady, Andreas F. Haag.

**Writing – original draft:** Rodrigo Ibarra-Chávez, José R. Penadés, Andreas F. Haag.

**Writing – review & editing:** Rodrigo Ibarra-Chávez, John Chen, José R. Penadés, Andreas F. Haag.

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
