## [Decision Letter · Decision Letter 0]

22 Jun 2021

Dear Jose,

Thank you very much for submitting your Research Article entitled 'Phage-inducible chromosomal islands promote genetic variability by blocking phage reproduction' to PLOS Genetics.

As you will see below, the reviewers' opinions on your manuscript are diverse. I invite you to revise your manuscript paying close attention to the reviewers' comments, especially considering the relevant objections raised by reviewers #1 and #3.

If you decide to revise the manuscript for further consideration at PLOS Genetics, please aim to resubmit within the next 60 days, unless it will take extra time to address the concerns of the reviewers, in which case we would appreciate an expected resubmission date by email to plosgenetics@plos.org.

[LINK]

We are sorry that we cannot be more positive about your manuscript at this stage. Please do not hesitate to contact us if you have any concerns or questions.

Yours sincerely,

Josep Casadesús

Section Editor: Prokaryotic Genetics

PLOS Genetics

Reviewer's Responses to Questions

**Comments to the Authors:**

Reviewer #1: Ibarra-Chavez et. al. present measuring the impact of phage-induced SaPI activity on rates of resistance marker acquisition in bacterial recipient cells. They observe increased transduction of resistance markers and decreased lysogeny of inducing phages when bacterial hosts encode SaPIs. From this, the authors conclude that SaPIs actively promote host DNA co-transduction to increase host population genetic diversity and thus increase survival. The work is presented in a concise manner and I appreciated the exploration of the potential importance a non-toxigenic SaPI. Unfortunately, I disagree with the conclusions the authors draw from the data. I find the work to be lacking in novelty and consideration of several major factors that are invalidating to the conclusions in the text (see major points below). I do not recommend publication of this work in any journal without serious reconsideration of the scope of the paper.

Major points

1. The manuscript (including the statement in lines 78-79) fails to acknowledge what is currently already known in the broad field of PICI biology regarding PICI-mediated interference with phages. Being familiar with the literature, it is unclear what new biological insights are obtained from this work.

2. Multiple figures in the text do not enrich current understanding of the SaPI system but rather reiterate known information with the tone of novelty.

a. Figure 2 shows that phage titer is decreased with SaPI activity. This phage repression activity has previously been explored in molecular detail (Ram et. al. PNAS 2014). The reduction in phage titer is an expected result from what is already known about the system

b. Figure 2 also shows increased cell viability in the presence of SaPIs, which was demonstrated in the original molecular characterization of the system (Ruzin et. al. Mol Micro 2001) and continues to be a theme in SaPI biology. Similarly, Figure 3 shows a decrease in lysogenization frequency in hosts with active SaPIs, another phenomenon described in the original molecular characterization of SaPIs by Ruzin et. al.

3. The data and resulting analysis confuse rather than clarify a fairly intuitive concept: if a host cell harbors a phage-interference mechanism like a SaPI, phage infection will be impaired resulting in decreased lysogenization (because fewer phage particles can be produced) and increased host population survival which translates to increased probability of bacterial DNA uptake by transduction. I would argue that any mechanism that impairs phage reproduction in a population could effectively result in the same phenomenon. Thus, there does not appear to be any new biological insights gleaned from this work.

4. The authors draw the conclusion that increased population diversity of recipient cells containing SaPIs acts as an ecological bet-hedging strategy to promote survival in stressful conditions. I do not believe that the data presented in this manuscript directly support this claim. Further assessment of the relative fitness of these populations is required in order to make such a claim about an ecological strategy.

Minor points

1. (Lines 44-48) It is unclear whether all surviving cells in these populations have acquired new DNA.

2. (Lines 54-55) The different types of transduction (generalized vs. specialized vs. lateral) need some clarifying definition in the text.

3. Line 87 the designation of the SaPI pT1028 is unclear; this nomenclature is appropriate for a plasmid. If the original publication (is there a citation for this SaPI?) erroneously referred to this element as a plasmid it should be corrected and not perpetuated in the literature

4. Lines 89-90 could be clarified by stating here (rather than later) that these phages are the inducing phages for the SaPIs. I would also like some clarification on cross-activity between the selected SaPI-phage pairs (i.e. can phi11 induce pT1028 and 80alpha induce SaPIbov1?)

5. A model of the experimental set-up would greatly aid the reader’s understanding of lines 80-101.

6. Consider referring to RN4220 and other strain numbers as something more understandable to the general reader (e.g. “susceptible host”)

7. Please refer to the specific sub-panel of figures referenced in the text (e.g. line 128 should specifically reference Figure 3A)

8. Line 77 wording requires clarification regarding PICI and host DNA co-transduction.

9. Line 130 and elsewhere, please detail the actual degree of reduction rather than just calling it statistically significant.

10. Lines 175-181 need editing, there are words missing from some of these sentences that are required for grammatical correctness and clarity, specifically “creating news that” and “since as mentioned the output”

11. Figures 1-4: please provide information about the limit of detection for these assays.

12. Figures 1, 2, 3B: One-way ANOVA statistical analysis is not appropriate for these data.

13. Figures 1B, 2B: It appears that there is still some SaPI activation in the absence of sri in 80alpha. This is not addressed in the text.

14. Figure 3: Experiments conducted in duplicate are compared to experiments conducted 5 times. This seems inappropriate considering nearly all other experiments were conducted in biological triplicate at minimum.

15. Figure 4: Individual data points on the bar graphs (as in Figures 1-3) are absent on this figure. This figure would also be strengthened by inclusion of a third biological replicate.

16. Figure 5 is extremely confusing and has many elements that are not factually correct. It could be clarified by removing the shading in the large squares, including the initial phage infection step to orient readers to the overall process, and some serious reconsidering of arrow colors. This figure also contains some misleading information, specifically in the right-side panel that shows an individual SaPI+ cell surviving phage infection (it does not in fact protect the individual cell) and successful transduction blocking further phage infection (also not supported by what is currently known). This figure should be seriously reconsidered.

Reviewer #2: Interesting manuscript pointing to a central role of PICI's in transduction and importantly also for lysogenization which could be highlighted more and discussed: Why do bacteria avoid becoming lysogens and rather carry PICIs?

Line 32: A more diverse - more than what? The statement of diversity is somewhat overstating the results of the manuscript. Basically, the diversity is obtained by cells being able to survive without being lysogens - not that a greater diversity of cells in general or transduced cells are being created. The abstract should put more focus on the fact that more cells are able to survive in a phage environment without becoming lysogens.

line 94: Why is RN4220 being used here? The strain seems in general to suffer from several problems so a strain not having undergone rounds of mutagenesis would be preferable

Figure 2, line 368: Are the viable cells transductants or just viable cells in culture?

Figure 3 and figure 4: You are not commenting on the results of figure 3B in the text and basically the data in figures 3 and 4 seem in part to be the same but presented in different ways.

line 177: new ones instead of news?

Reviewer #3: The manuscript by Ibarra-Chávez et al, entitled "Phage-inducible chromosomal islands promote genetic variability by blocking phage reproduction" describes a new role for PICIs as promoters of phage-driven gene transfer. The data provided for two different PICIs, induced in the presence of two different phages in Staphylococcus aureus are convincing and overall support the model provided in Fig. 5. The main caveat of the manuscript is that it requires extensive prior knowledge of the PICIs biology and bibliography. The introduction is rather minimalist and does not provide sufficient information to guide the reader toward the initial hypothesis. This version needs improvement to reach the large audience and scope of the journal.

L.62: The description and biology of PICIs is rather minimalist, and not really adapted to a large audience. In particular, the phage-encoded inducer needs additional description as it is needed to understand Fig. 1.

L. 75, What is meant by “local population?”, if the model considers mixed or non-homogenous populations, it could be further explained.

L.79, it is not clear from the abstract or introduction how the work provided here relates to bacterial (genome) evolution.

L. 84-88: Two different SAPIs and cognate phages were used in this study for the sake of generalization (as indicated l. 96). Although the use of SapIbov1 is clearly explained, it is hard to get why pT1028 was chosen? A more detailed justification and the addition of references for these two SAPIs models would help the reader. In addition, as the authors study many different PICIs models, why not including different strains and classes of PICIs from different bacterial strains in the study to generalize even more?

L.93: Generalized and lateral transduction should be defined in the introduction to understand the subtility of using 2 different reporters.

L. 94: Strain RN4220 is a restriction-defective derivative of RN450, as indicated in Table 1. Why such a strain is needed. Table 1 could also mention the genotype of the original RN450.

L.97, is there a reference to provide for pT1028 induction by 80α?

L.99, the hypothesis mentioned could be rephrased to help the reader.

L. 110, it is hard to understand the basis of such hypothesis without additional explanation. If the phage titer decreases, is it because each cell is less productive, or because cells become less susceptible?

L. 117, the link between lysogeny and transduction facilitation is not obvious as stated.

L. 156, in this experiment the genetic variability observed is driven by the co-occurrence, and maintenance in the overall population, of several type of cells. Does it influence the population heterogenicity on the long term? How were the time of plating (18 h post-infection) chosen? In other words, is it possible to obtain a different distribution depending on the incubation time?

Discussion: Could the authors precise why they mention lysogenisation is a stochastic process and provide the appropriate references?

As stated in the text, the heterogenicity of the population driven by SAPIs could provide an advantage to the overall population. This could be tested experimentally.

Additional minor points: Phage titer instead of “phage titre”, several occurrences. Strain RN450 is missing in Table 1.

**Have all data underlying the figures and results presented in the manuscript been provided?**

Reviewer #1: Yes

Reviewer #2: **No: **Figure 4 represents relative population frequency and the actual counted number of cells are not provided

Reviewer #3: Yes

PLOS authors have the option to publish the peer review history of their article (what does this mean?). If published, this will include your full peer review and any attached files.

Reviewer #1: No

Reviewer #2: No

Reviewer #3: No

---

## [Decision Letter · Decision Letter 1]

21 Sep 2021

Dear Jose,

Thank you very much for submitting your Research Article entitled 'Phage-inducible chromosomal islands promote genetic variability by blocking phage reproduction' to PLOS Genetics.

As you will see below, reviewer #1 expresses disappointment about the fact that major concerns cited in the first round of review have not been addressed. I must agree with their view. Reviewer #3 also expresses a similar opinion in comments to the editor (but their comments to the authors are certainly more positive). After pondering the situation, I think that the right decision is to give you a second opportunity to address the comments made by reviewer #1. Please note that dismissal of criticism should not be customary.

If you decide to revise the manuscript for further consideration at PLOS Genetics, please aim to resubmit within the next 60 days, unless it will take extra time to address the concerns of the reviewers, in which case we would appreciate an expected resubmission date by email to plosgenetics@plos.org.

[LINK]

We are sorry that we cannot be more positive about your manuscript at this stage. Please do not hesitate to contact us if you have any concerns or questions.

Yours sincerely,

Josep Casadesús

Section Editor: Prokaryotic Genetics

PLOS Genetics

Reviewer's Responses to Questions

**Comments to the Authors:**

Reviewer #1: The manuscript re-submitted by Ibarra-Chavez et. al. includes positive revisions increasing the amount of background information in the introduction and substantial figure revisions that have made the data more aesthetically pleasing. However, the authors failed to address the major concerns cited in the first round of review. Rather than constructively responding to the critiques raised, the authors chose to dismiss nearly every major critique I provided with little to no attempt to address the logic or concern itself. The authors instead chose to re-state their original hypotheses and decline to include any additional experiments or controls that would strengthen the claims they are trying to make. The manuscript continues to fail to acknowledge what is currently already known in other PICI systems where a role in protecting bacteria from phage predation has already been detailed. It is not to say that there are not additional experiments to be done in this area, there clearly are, however if a reader new to this field picked up this paper, they would assume based on how it is written that PICIs interfering with phages has been shown only in this system and here for the first time. Rather than continue to provide detailed feedback on this manuscript I will simply re-iterate a few crucial details that are particularly troubling:

1. The model in figure 5 has undergone edits and is still extremely misleading and unclear. The authors conduct experiments at MOI 1:10 (phage:bacteria) and do not see a change in cell viability in SaPI+ hosts, which they have interpreted here as SaPI+ cells surviving phage infection at the single-cell level. This data simply cannot be interpreted as survival or SaPI-mediated protection at the single-cell level. At the very least a high MOI infection should be performed to test this hypothesis directly and provide actual supporting data. It is also extremely unclear from the model that transduction results from an initial round of phage infection.

2. The SaPINY940 data is not explained satisfactorily in the text. SaPINY940 is stated to be induced by sri but the citation for this information (Ref. 26 Tormo-Mas et. al. Nature 2010) does not include this SaPI in its analysis. It is unclear how the authors conclude that sri is the inducer and unclear if there is SaPI induction in the absence of sri. From the data in this manuscript, it appears there is some level of induction without the inducer, which makes this SaPI a questionable choice to include in this analysis attempting to connect co-transduction with SaPI activation. If the authors wish to suggest that SaPINY940 has some additional phage resistance capacity outside direct phage parasitism, this should be explored more thoroughly, especially considering the conceptual similarity to recent discoveries in other satellite-encoded defense systems (e.g. Bikard Lab https://www.biorxiv.org/content/10.1101/2021.01.21.427644v1). I expect the authors to conclude such investigation is outside of the scope of the current manuscript, I can agree with that provided that the net result of such protection (at the single cell level) is not included in the model (figure 5).

3. I do not agree that the authors have discovered a novel role for SaPIs (or PICI’s in general) in diversity-generating transduction. The authors do not include controls to convince readers that SaPIs specifically (and not any phage defense in general) can have this effect.

4. I disagreed with the author’s conclusion that SaPIs promote host DNA co-transduction to increase host population genetic diversity. The authors responded “our main conclusion is not that SaPIs are actively promoting host DNA co-transduction but rather that SaPIs promote the survival of transductants by protecting the overall population and potentially individual cells from phage predation”. If that is the case the authors should more carefully state their claims, for example, the authors have chosen to title Figure 1 “SaPIs promote increased levels of horizontal DNA transfer.” These details are in direct conflict with the statement in the authors’ rebuttal.

5. A note about statistics: I am not questioning general significance of the differences being claimed, the averages are obviously different by eye. However, ANOVA analysis with post hoc EMM is not the most appropriate method for analyzing these data sets. Every comparison made in this manuscript is between 2 well-defined groups that do not appear to have similar distributions. This type of data is best analyzed by Student t-test, which does not assume that your data are distributed the same in all sets. ANOVA assumes all sets have equal distribution and attempts to find relevant pairs of groups to analyze. The experimental set-up here defines these groups so this is not necessary. A single t-test per group results in the lowest probability of making a Type I error. Choosing ANOVA and then post hoc analysis may “work”, but it increases the risk of Type 1 error, artificially inflates the p-value, and is an unnecessarily complicated analysis for the type of data being presented in this manuscript.

Reviewer #2: I am happy with the response

Reviewer #3: Overall, the text of the manuscript has improved.

Title could be completed: Phage-inducible chromosomal islands promote genetic variability by blocking

phage reproduction and protecting the transductants from phage lysis

Line 41: “of the general bacterial population”, this is not obvious what is meant by general population, at the population level ?

**Have all data underlying the figures and results presented in the manuscript been provided?**

Reviewer #1: Yes

Reviewer #2: Yes

Reviewer #3: None

PLOS authors have the option to publish the peer review history of their article (what does this mean?). If published, this will include your full peer review and any attached files.

Reviewer #1: No

Reviewer #2: No

Reviewer #3: No

---

## [Decision Letter · Decision Letter 2]

1 Mar 2022

Dear Jose,

Thank you very much for submitting a revised version of your manuscript 'Phage-inducible chromosomal islands promote genetic variability by blocking phage reproduction and protecting transductants from phage lysis' to PLOS Genetics. As you will see below, the manuscript has been examined by one reviewer (chosen from those who had evaluated previous versions). Based on the reviewer's comments, I think that an additional round of review might improve the story. Therefore, I invite you to consider the reviewer's objections. Then you may either modify the manuscript or explain why you disagree with the reviewer if such is the case.

[LINK]

Please let me know if you have any questions while making these revisions.

Yours sincerely,

Josep Casadesús

Section Editor: Prokaryotic Genetics

PLOS Genetics

Reviewer's Responses to Questions

**Comments to the Authors:**

Reviewer #1: The manuscript is much improved from the initial submission, but there are some points on which it appears we will have to agree to disagree. The experiments that were used to conclude that "SaPIs protect individual cells from phage attack" are inappropriate and do not make any sense to me. At an MOI=1, ~63% of cells are expected to be infected, unless the authors have a compelling reason to indicate the Poisson distribution does not apply to the system under study.

The authors state: "We reasoned that if SaPI-mediated protection acted at a population level alone, we should not observe any differences in cell viability, phage, or transduction titers between SaPI-positive and SaPI-negative recipient cells under these conditions. "

I disagree. Seeing as nearly 40% of the cells are uninfected, and phage titers are reduced following an initial round of infection by the SaPI it is no wonder the authors have a difference in cell viability. This does not support the model that SaPIs protect individual cells from phage attack. At an MOI =5, 99% of cells would be infected and I am certain that the authors would find absolutely no difference in cell survival +/- SaPI, but that is not the experiment they chose to perform.

**Have all data underlying the figures and results presented in the manuscript been provided?**

Reviewer #1: Yes

PLOS authors have the option to publish the peer review history of their article (what does this mean?). If published, this will include your full peer review and any attached files.

Reviewer #1: No

---

## [Editor Report · Decision Letter 3]

14 Mar 2022

Dear Jose,

I am pleased to inform you that your manuscript entitled "Phage-inducible chromosomal islands promote genetic variability by blocking phage reproduction and protecting transductants from phage lysis" has been editorially accepted for publication in PLOS Genetics. Congratulations!

Yours sincerely,

Josep Casadesús

Section Editor: Prokaryotic Genetics

PLOS Genetics

Comments from the reviewers (if applicable):

**Data Deposition**

http://datadryad.org/submit?journalID=pgenetics&manu=PGENETICS-D-21-00720R3

**Press Queries**

---

## [Editor Report · Acceptance letter]

25 Mar 2022

PGENETICS-D-21-00720R3 

Phage-inducible chromosomal islands promote genetic variability by blocking phage reproduction and protecting transductants from phage lysis 

Dear Dr Penadés, 

We are pleased to inform you that your manuscript entitled "Phage-inducible chromosomal islands promote genetic variability by blocking phage reproduction and protecting transductants from phage lysis" has been formally accepted for publication in PLOS Genetics! Your manuscript is now with our production department and you will be notified of the publication date in due course.

With kind regards,

Zsofia Freund

PLOS Genetics

On behalf of:
